# Occurrence and backtracking of microplastic mass loads including tire wear particles in northern Atlantic air

Isabel Goßmann [1,2], Dorte Herzke [3,4], Andreas Held[5], Janina Schulz[1], Vladimir Nikiforov[3], Christoph Georgi[5], Nikolaos Evangeliou [6], Sabine Eckhardt [6], Gunnar Gerdts[7], Oliver Wurl[1,2] & Barbara M. Scholz-Böttcher [1] ✉

Few studies report the occurrence of microplastics (MP), including tire wear particles (TWP) in the marine atmosphere, and little data is available regarding their size or sources. Here we present active air sampling devices (low- and high-volume samplers) for the evaluation of composition and MP mass loads in the marine atmosphere. Air was sampled during a research cruise along the Norwegian coast up to Bear Island. Samples were analyzed with pyrolysis-gas chromatography-mass spectrometry, generating a mass-based data set for MP in the marine atmosphere. Here we show the ubiquity of MP, even in remote Arctic areas with concentrations up to 37.5 ng m$^{-3}$. Cluster of polyethylene terephthalate (max. 1.5 ng m$^{-3}$) were universally present. TWP (max. 35 ng m$^{-3}$) and cluster of polystyrene, polypropylene, and polyurethane (max. 1.1 ng m$^{-3}$) were also detected. Atmospheric transport and dispersion models, suggested the introduction of MP into the marine atmosphere equally from sea- and land-based emissions, transforming the ocean from a sink into a source for MP.

Microplastics (MP) in the air have been of increasing interest lately. Studies documenting these small plastic particles within the size range of 1 μm to 5 mm in the atmosphere are emerging[1–5]. Still, atmospheric MP transport and their ocean-atmosphere fluxes are widely unknown[6]. Simulations of atmospheric pathways suggest that notable amounts of terrestrial micro- and nanoplastics are being transported via the atmosphere to the marine environment[6–8]. One study focused on the atmospheric transport of tire wear particles (TWP) calculated that TWP in the PM2.5 (aerodynamic diameter ≤2.5 μm) size range dispersed more extensively in the atmosphere than larger particles (PM10; aerodynamic diameter <10 μm). While larger TWP (incl. PM10) deposited rather near hotspot emission regions, particles in the PM2.5 size

range even reached the polar regions and are therefore, a highly relevant size range when analyzing marine atmospheric MP contamination[8]. A recent study by ref. 9 suggested that MP stay not only in the ocean but might also be re-emitted into the marine atmosphere via sea spray and bubble bursting. Two more studies stated that oceanic re-emissions of MP were believed to be the most important source of atmospheric MP, showing annual global emissions >0.8 million metric tons[7,10].

All types of particles, fibers, and fragments of synthetic origin, abraded polymer-based paint flakes and TWP, are by now included in the definition of MP[11]. The most used polymers are the thermoplastic polymers polyethylene (PE), polypropylene (PP), polystyrene (PS),

[1]Institute for Chemistry and Biology of the Marine Environment (ICBM), Carl von Ossietzky University of Oldenburg, P.O. Box 2503, 26111 Oldenburg, Germany. [2]Center for Marine Sensors, Institute for Chemistry and Biology of the Marine Environment (ICBM), Carl von Ossietzky University of Oldenburg, 26382 Wilhelmshaven, Germany. [3]NILU - Norwegian Institute for Air Research, The FRAM Centre, P.O. Box 6606, Langnes, 9296 Tromsø, Norway. [4]NIPH – Norwegian Institute for Public Health, P.O.Box 222 Skøyen,, 0213 Oslo, Norway. [5]Chair of Environmental Chemistry and Air Research, Technische Universität Berlin, 10623 Berlin, Germany. [6]NILU - Norwegian Institute for Air Research, Instituttveien 18, 2007 Kjeller, Norway. [7]Alfred Wegener Institute, Helmholtz Center for Polar and Marine Research, 27483 Heligoland, Germany. ✉e-mail: bsb@icbm.de

poly(ethylene terephthalate) (PET), poly(vinyl chloride) (PVC), polycarbonate (PC), poly(methyl methacrylate) (PMMA), and polyamide (PA6). These basic polymers are also the ones mostly detected in the atmosphere and the environment in general. Together with polyurethanes (PUR), which are also in high demand and frequently found in the environment[4,12–18], these polymers comprise ~80% of the European plastic demand[19]. TWP are released from tire tread, which is manufactured from natural and synthetic elastomeric rubber[20,21].

In general, studies analyzing MP in the atmosphere are scarce, especially when it comes to the marine atmosphere and pathways of atmospheric transport into the marine environment. A comprehensive review[6] summarized six publications, which are displayed in the supplementary information SI; (Table S1) together with the latest study by ref. 22. The presented data were restricted to particle number-based concentrations analyzed with Fourier-transform infrared spectroscopy (FTIR) or μ-Raman spectroscopy. In the <200 μm size range, MP counts ranged up to 85 particles m$^{-3}$ (refs. 23–26), with some particles as small as 5–10 μm identified[25].

Atmospheric MP studies are often based on passive sample collection. Passive sampling benefits from easy usability, low acquisition costs, and no necessity for power at the sampling site[27,28]. When sampling in the marine environment, typically on a research vessel, the sampling time is limited. Hence, passive sampling is not a realistic option. Active air sampling using a vacuum pump provides an effective and replicable sampling method for the analysis of suspended atmospheric contaminants with well-defined sampling volumes and periods[27,28].

The analysis of MP was performed according to refs. 21,29 using pyrolysis-gas chromatography-mass spectrometry (Py-GC-MS), which enabled simultaneous trace identification and quantification of MP including TWP. As already presented in detail in earlier studies[4,17,30,31], polymers were quantified as clusters (indicated by prefix C-), referring to defined base polymers with characteristic thermally generated building blocks[4,17,31,32]. The applied method included the common plastic polymer clusters, C-PE, C-PP, C-PS, C-PET, C-PVC, C-PMMA, C-PA6, C-MDI-PUR, C-PC, and CTT (car tire tread) as well as TTT (truck tire tread).

In this study, we present quantitative data for MP mass loads including TWP in northern Atlantic air along a south/north gradient that included the Arctic. Two different active sampling techniques were applied, compared, and evaluated. The aim was to get insights into MP composition, distribution, and especially sources of (marine) atmospheric MP contamination. Challenges regarding the determination of MP in the marine atmosphere were highlighted.

## Results and discussion
### Operational ship blanks and exclusion of selected polymer clusters

To minimize secondary contamination from the ship in advance, the sampling devices were positioned at elevated sites at the ship's bow and sampling was strictly restricted to steaming phases only. However, operational blanks for air sampling were taken throughout the entire sampling period to monitor secondary contamination released from the ship's environment or the two different sampling devices. Low-volume (LV, 54–417 m$^3$ air per sample) samplers were pre-assembled beforehand under a laboratory clean bench. In contrast, high-volume (HV, 288 to 2184 m$^3$ air per sample) samplers were regularly opened on board to exchange the aluminum rings used as sampling targets. This resulted in different compositions and mass-loads for the respective operational blanks (Fig. 1). In the following, the polymer clusters found in the respective operational blanks are discussed and their secondary contamination with its potential effects on further quantification is evaluated.

The indicator ion for C-PMMA appeared almost ubiquitous in the seven transects (T1–T7) in both samplers (LV and HV) and all

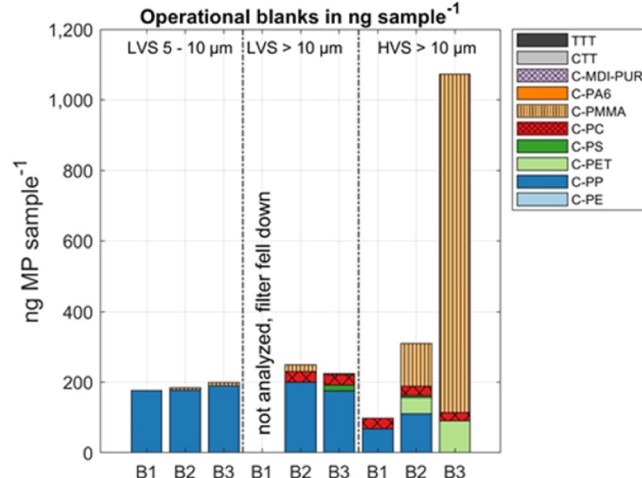

**Fig. 1 | Microplastic (MP) concentrations in operational blanks taken over the sampling period.** MP concentration in the three operational blanks B1 (between T1 and T2), B2 (between T5 and T6) and B3 (after T7) in ng sample$^{-1}$ for the low-volume sampler (LVS), in the size ranges 5–10 μm and >10 μm, and the high-volume sampler (HVS).

investigated size fractions (>10 μm for LV & HV samples; 5–10 μm for LV samples). Marine coatings are amongst others a plausible source of C-PMMA[31]. On the R/V Heincke polishing and painting the ship was a daily routine for the crew. This might be mirrored in the operational blanks, particularly in the elevated C-PMMA contents of the HV samplers, where direct contact with the ship environment was inevitable (Fig. 1). C-PMMA concentrations in the air were unquestionably of anthropogenic origin. However, differentiation of the high C-PMMA load between northern Atlantic air (and ships in general) and from the ship, as the sampling platform, was impossible. Accordingly, we decided to exclude C-PMMA from further discussion.

The >10 μm size fraction of both samplers (LV and HV) contained C-PC in all samples and operational blanks (Fig. 1) in comparable concentrations of ~30 ng sample$^{-1}$ or operational blank$^{-1}$. Again, the origin of this polymer cluster in the >10 μm size fraction could not be exclusively assigned to the northern Atlantic air, but more likely to ship work related to epoxide coatings[31] and to the samplers themselves. This led to the exclusion of C-PC in the >10 μm size fraction.

Both LV sample size fractions (5–10 μm and >10 μm) displayed an almost constant C-PP value for samples and operational blanks (~200 ng sample$^{-1}$). Here, a sampler-related C-PP content was assumed and led to the exclusion of C-PP from LV sampler results.

C-PS was occasionally detected in both samplers, as well as C-PP and C-PET for the HV sampler, but did not show any operational blank-related pattern and thus, did not suggest an overall secondary contamination during sampling or sample preparation. Therefore, these polymer clusters were not excluded from the discussion. As the operational blanks indicated possible contamination related to the sampler preparation but not the entire sampling process, a blank subtraction of the respective data was not conducted. Instead, these polymer clusters are marked (Δ) in the respective figures and $^\Delta$ in the text to indicate that the concentration might be partly impaired by secondary contamination. In addition, Figs. S1 and S2 in the supplementary information show the absolute mass loads (ng sample$^{-1}$) of each polymer cluster next to the operational blanks for both samplers.

### Low-volume samples

After the exclusion of selected polymer clusters, MP were still present in all samples of the size fraction >10 μm. In the 5–10 μm size fraction, five out of seven samples contained MP (Fig. 2b). Total mass loads ranged from <limit of quantification (LOQ) to 1.82 ng MP m$^{-3}$. Limits of

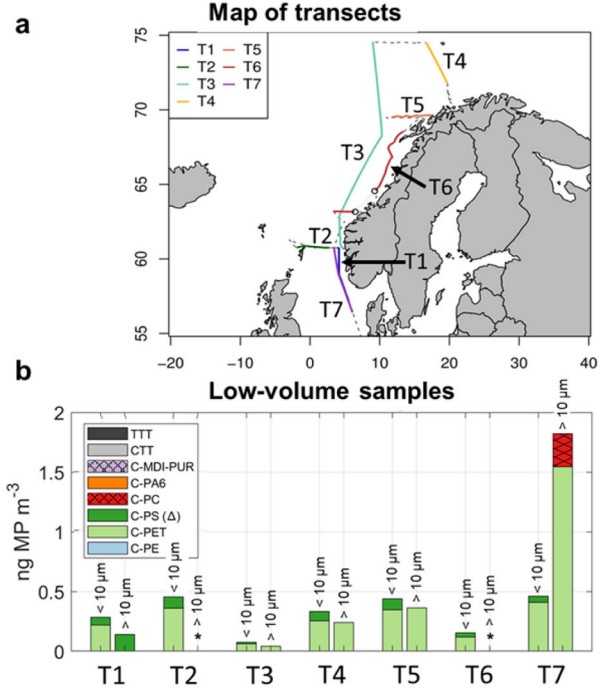

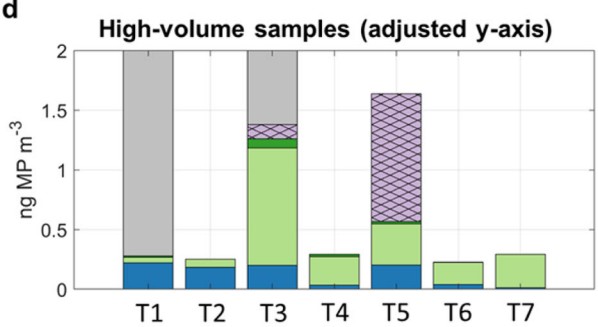

**Fig. 2 | Total airborne microplastic (MP) concentrations in ng m$^{-3}$ for the low-volume (LV) and high-volume (HV) samplers for the T1–T7 samples. a** Map of transects T1–T7 of air sampling during the HE578 cruise, June 2021. Map data is from Wickham H (2016). ggplot2: Elegant Graphics for Data Analysis. Springer-Verlag New York. ISBN 978- 3-319-24277-4, https://ggplot2.tidyverse.org. **b** LV samples in the >10 μm and <10 μm (equals 5–10 μm) size fractions. **c** HV samples in >10 μm size fraction. **d** HV samples with an adjusted *y*-axis (0–2 ng MP m$^{-3}$). * = concentration <limit of quantification (LOQ); Δ = concentration of polymer clusters might be influenced by secondary contamination.

detection (LOD) and LOQ are displayed in the supplementary information SI, (Table S2).

Irrespective of size fraction, C-PET was the dominant polymer cluster (max. 1.54 ng m$^{-3}$). C-PS$^{Δ}$ also appeared frequently but in much lower concentrations (max. 0.14 ng m$^{-3}$). C-PC (5 – 10 μm fraction only) was quantified once with a concentration of 0.28 ng m$^{-3}$ (T7). With one exception (T7), the MP concentrations were higher in the size fraction >10 μm than in the 5–10 μm size fraction. Detailed polymer cluster data is available in the supplementary information SI, (Table S3 and S4)

### High-volume samples
All HV samples contained MP. The summed concentrations ranged from 0.23 to 37.5 ng MP m$^{-3}$. Clear evidence for CTT (Fig. 2c) was obtained in T1 (35.3 ng m$^{-3}$) and T3 (13.2 ng m$^{-3}$), representing 94% and 87% of the total MP concentration in the respective samples. However, by adjusting the *y*-axis to the same scale as used for the LV samples (Fig. 2d), concentrations of other polymer clusters became evident including C-PP$^{Δ}$, C-PET$^{Δ}$, C-PS$^{Δ}$, and C-MDI-PUR. These were present in much lower amounts. When TWP were excluded (both CTT and TTT), the summed concentrations never exceeded 2 ng m$^{-3}$ and represented the same order of magnitude as what was observed in the LV samples (>10 μm). Arranged in descending order, the polymer concentrations were C-PET$^{Δ}$ > C-PP$^{Δ}$ > C-MDI-PUR > C-PS$^{Δ}$, resulting in mean relative percentages of 56% (C-PET$^{Δ}$), followed by 31% (C-PP$^{Δ}$), 11% (C-MDI-PUR), and 3% (C-PS$^{Δ}$).

### MP composition and comparison with literature data
Due to the limited availability of literature data concerning MP in the marine atmosphere, the scope of comparison is limited. An additional challenge was the variation in reported sampling and analytical methods, as well as sampling sites. The seven published, particle-number-based studies mentioned in the introduction and the supplementary information SI, (Table S1) did not include TWP. Therefore,

TWP are neither taken into account in the following comparison, nor are the derived relative proportions listed. These proportions were calculated to enable an approximate comparison of the particle–number-based studies with our particle-mass-based study. Five out of seven studies reported PET or polyester as the main detected polymer type, with occurrences between 29 and 56%[23,24,26,33,34]. This is in accordance with both the LV and HV samples of this study, where C-PET had by far the largest proportions with 67% and 56%, respectively. The second most dominant polymer cluster for the LV samples was C-PS$^{Δ}$ (17%), for which there is only limited agreement in the literature. Only[25] highlighted PS as a dominant polymer over PP and PE, but did not present any percentages, whereas two other studies reported only small contributions from PS (10%[23] and 6%[33]). The dominant and ubiquitous presence of C-PS in the LV samples of this study might be related to the very low limit of detection (LOD, -1 ng; SI, Table S8) using Py-GC/MS, which facilitates easier identification of trace C-PS$^{Δ}$ concentrations. Other polymer clusters might have also been present in the LV samples, but due to higher LODs of some polymers (e.g., CTT), they might have evaded identification in the comparably low sample volumes. This hypothesis is confirmed by the HV samples of this study. Due to the overall higher sample volume, a greater polymer diversity was detected. In these samples, C-PS$^{Δ}$ occurred frequently, but its proportion was low (around 3%). In the HV samples, C-PP$^{Δ}$ was the second most abundant polymer cluster, with an average share of 31%. PP also occurred frequently in other studies but with lower relative proportions, ranging from 7 to 22%[23,25,33,34]. In two transects (T3 and T5), C-MDI-PUR was a prominent component (average 11%) detected via HV sampling. Only one study reported the detection of PUR in marine air samples with a 5% contribution[24]. However, the most prominent polymer clusters identified are largely in agreement with the sparsely available literature data and underline the predominance of C-PET, in particular, in the marine atmosphere. In general, the comparison of particle- and mass-related MP results is limited and requires widespread harmonization[35].

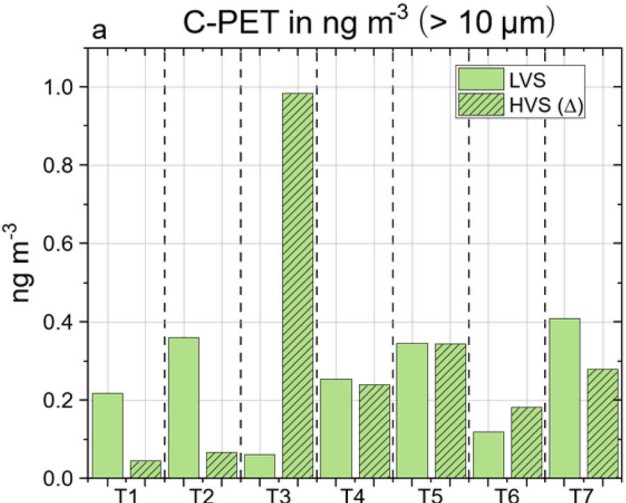

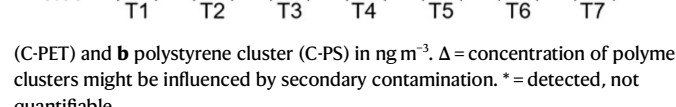

**Fig. 3 | Comparison of both samplers (low-volume (LVS) and high-volume (HVS) sampler) based on the results of selected polymer clusters across the seven transects (T1–T7) for the >10 µm fraction. a** Polyethylene terephthalate cluster (C-PET) and **b** polystyrene cluster (C-PS) in ng m$^{-3}$. Δ = concentration of polymer clusters might be influenced by secondary contamination. * = detected, not quantifiable.

## Comparison of LV and HV samplers

For HV samples only the fraction >10 µm was available for MP analysis. Accordingly, the comparison between the two different sampling approaches was limited to this size fraction. Unfortunately, some of the polymer clusters had to be excluded from the discussion due to their occurrence in the operational blanks. Therefore, the two sampling techniques could only be compared concerning the polymer clusters, C-PET and C-PS (Fig. 3a, b).

Irrespective of the sampling technique, the concentration of C-PET ranged between 0.05 and 0.41 ng m$^{-3}$, except for the HV sample in T3 (0.98 ng m$^{-3}$). The detected C-PET mass loads ranged on the same order of magnitude and T4, T5, T6, and T7 displayed an especially strong resemblance. The unusually high mass loads of C-PET in the HV sample of T3 could be explained by the presence of visible fiber accumulations in the sample (SI, -picture of filter cake in Fig. S3). C-PS$^\Delta$ concentrations seemed to vary among all samples, but overall, the mass loads of C-PS$^\Delta$ were within the same order of magnitude and never exceeded 0.1 ng m$^{-3}$.

Even though direct comparison of the LV and HV samplers was highly restricted, both showed promising similarities, which underlined their suitability for air sampling on board in general. Each sampler had its advantages and disadvantages. Low sample volumes reflected a putatively limited diversity in polymer types, but in the case of the LV sampler, it was more resistant to secondary contamination emerging directly from the sampling site (ship). Furthermore, it was simple to prepare the LV samplers in advance and easy to handle, both on board and in the laboratory. In contrast, the elevated air volumes of the HV sampler revealed a greater polymer cluster diversity in the air. Larger sample volumes often ensure a more reliable analysis of polymers with higher LOD and LOQ, which might directly result in a higher polymer cluster diversity for the HV samples compared to the LV samples. The aluminum rings, acting as sample collectors, had to be inserted into the sampler on board and replaced after each sampling procedure. Accordingly, this sampling technique is more vulnerable to any secondary contamination. In addition, the aluminum rings were, due to their size, less convenient to handle in the laboratory. However, both sampling techniques have the potential to give valuable insights into the MP composition of marine air. In particular, their combination and extension to include different size fractions is certainly a suitable approach for gaining more detailed insights into MP concentrations in the atmosphere.

## MP distribution and sources

To discover potential MP sources in the marine atmosphere, we used the Hybrid Single-Particle Lagrangian Integrated Trajectory (HYSPLIT) and the FLEXible PARTicle (FLEXPART) dispersion models to obtain information about the origin of air masses, which arrived to the ship and were hence sampled. Despite having a general idea of the possible polymer origins, the resulting back trajectories (HYSPLIT) and emission footprints (FLEXPART) should provide valuable, additional information about the anthropogenic impact of air masses, and therefore, particles. In Fig. 4a, b, selected parameters of both models are displayed. The modeling was conducted using various approaches, which are presented in the supplementary information SI, (Fig. S4).

Both models showed great similarity in terms of their calculated air mass origin, despite varying in terms of calculation and defined parameters (Fig. 4a, b). According to the FLEXPART model (Fig. 4a), the footprint emission sensitivities were the highest in oceanic (T1, T3, T5) and high Arctic regions (T4) showing that air arrived almost exclusively from marine areas. The same held true for the back trajectories of the HYSPLIT model (Fig. 4b). On the other hand, we sampled air masses, that had passed over the main land within the modeled time frames at T2, T6, and T7.

The highest total MP concentrations were found in the T1 and T3 samples. These samples showed particularly high TWP loads. According to Fig. 4a, b, large parts of the air masses that influenced transect T3 did not pass over any landmasses, raising the question of potential TWP sources. Recently, studies are emerging that propose sea spray as a transport vector and accordingly, a potential secondary source for MP in the marine atmosphere through remobilization from breaking waves causing bubbles of trapped air to rise and burst[7,9,10]. TWP has been already described to occur in the marine environment[21]. Recent own results from an unpublished study showed a significant accumulation of TWP in the sea surface microlayer of coastal marine waters. Hence, it might act as a potential re-emission source for TWP in the atmosphere entrained through sea spray, even without an anthropogenic source in the immediate vicinity. The same holds true for the TWP mass loads in T1, where the back trajectory of air masses and emission footprint suggested only marginal land contact (Fig. 4a, b). A study by[36] described a potential 6th oceanic plastic gyre in the Arctic, underlining the potential for re-emission of MP including TWP into the marine environment. Furthermore, TWP might remain in and travel through the atmosphere for longer periods of time, which has also been suspected in literature[8,37].

# Atmospheric transport and dispersion models

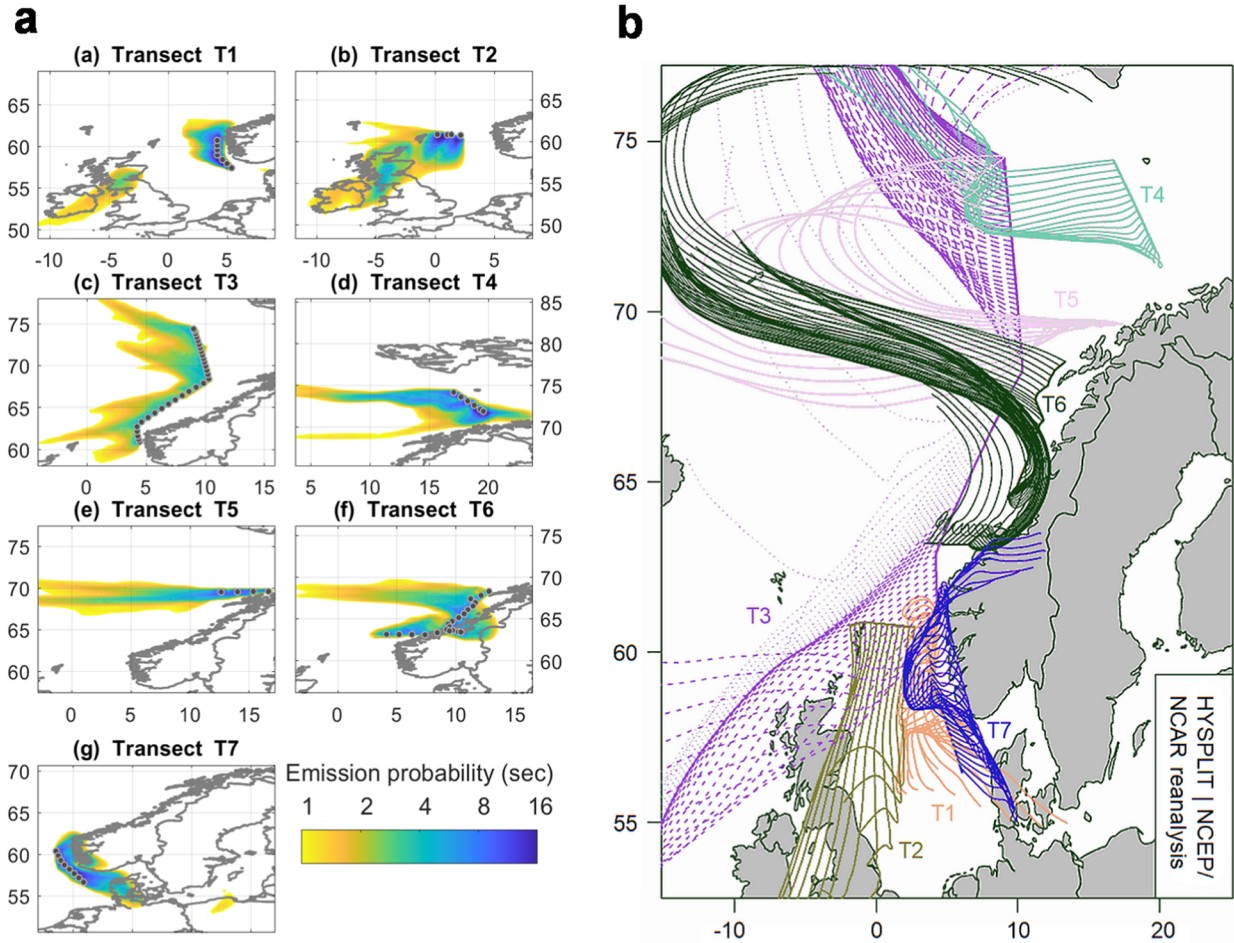

**Fig. 4 | FLEXPART FLEXible PARTicle dispersion model) and HYSPLIT (Hybrid Single-Particle Lagrangian Integrated Trajectory) model results for evaluation of particle and air mass origin for the respective transects, T1–T7. a** FLEXPART footprints simulating emissions of microplastic (MP) >10 μm size fraction at heights from 0–100 m above sea level for a duration of 30 days. **b** HYSPLIT back trajectories for the height of 30 m above sea level for a 72-h duration. Map data from Wickham H (2016). ggplot2: Elegant Graphics for Data Analysis. Springer-Verlag New York. ISBN 978- 3-319-24277-4, https://ggplot2.tidyverse.org.

The HV samples at T3 and T5 had the highest total MP mass loads, when disregarding TWP. For the LV samples, T7 stood out. The back trajectory of air masses indicated that T7 was clearly influenced by the southern Norwegian mainland. Several coastal cities provide a plausible source of C-PET, the main pollutant, which is most likely derived from synthetic fibers traveling through the atmosphere to the marine environment[37]. The models showed that T3 and T5 indicated only scarce (T3) or no (T5) influence from land. In these samples, the C-MDI-PUR was a dominant polymer cluster. According to ref. 31 ship coatings often contain epoxy and polyurethane coatings. Thus, the C-MDI-PUR loads in those samples might point to their re-emission via sea spray and atmospheric transport.

Overall, our findings are in agreement with and confirm that both atmospheric transport from land and oceanic re-emissions are important sources for atmospheric microplastics[6,7,9,10]. As the Arctic north Atlantic is considered to be the 6th oceanic plastic gyre[36] and an accumulation of MP is found in northern waters[38], a plausible source for MP is re-emission via sea spray. In consequence, the ocean, which was previously seen as an exclusive sink for MP, also acts as a source for atmospheric MP.

For further studies, the distribution of different size fractions should be a focus to examine the sources and transport routes including re-emission of MP from the ocean to the marine atmosphere. Potential associated risks for the marine biosphere and environment need to be more thoroughly investigated. A more standardized sampling combined with preferably large sample volumes and data analysis procedure would be helpful and would allow for better comparisons within the literature and a better understanding of atmospheric MP.

## Methods

### Sampling

Sampling was conducted during the HE578 cruise on the R/V Heincke in July 2021. The cruise track comprised of seven transects (T1–T7, Fig. 2a), which varied in length and total sampling time (12–91 h), along the Norwegian coastline to Bear Island (T1–T3) and back (T4–T7). Detailed information on sampling positions and time are described in the supplementary information SI, (Tables S5 and S6).

Air samples were taken with two different active sampling devices leading to a different total sampling volume for identical transects and sampling periods. The Norwegian Institute for Air Research (NILU) provided LV air samplers equipped with a filtration cascade to define particles in the size fractions >10 μm and 5–10 μm (stainless-steel mesh with pore size 10 μm and 5 μm, Haver & Boecker

OHG, Germany). The filter holders were made entirely of aluminum and were a replica of the NILU 2-stage filter holder, open face, #9639. Sampling volumes ranged from 54 to 417 m³. Prior to deployment, the filter holders and filters were kept at 450 °C for 8 h to remove all residual MP, then mounted and packed in a laminar flow fume hood. Technische Universität Berlin (TUB) mounted two HV samplers (DIGITEL Aerosol Sampler DHA-80, DIGITEL Elektronik AG, Switzerland) on the vessel. In addition to the standard PM10 collection, particles with an aerodynamic diameter >10 µm were deposited inside the sampler around the PM10 inlet on pre-cleaned aluminum rings (∅ 30 cm) placed below the 10 inlet jets. Total sampling volumes varied between 288 and 2184 m³. The identical HV samplers were placed next to each other to create duplicate measurements per transect. The discussion is based on the mean results of the duplicates. Pictures of the air samplers and set-up are provided in the supplementary information SI, (Fig. S5).

Air samplers were mounted on the observation deck of R/V Heincke at approximately 12 meters above sea level on the bow of the vessel. Regular research operations were not performed on the observation deck. MP sampling was only conducted while the ship was steaming to avoid any ship-related secondary MP and ship exhaust contamination to the greatest possible extent. Therefore, a predominantly undisturbed and uncontaminated sampling environment was expected.

Operational blanks of both sampling techniques were prepared during the research cruise to get an impression of possible secondary contamination during preparation of the samplers and handling on board. For this purpose, the respective sampler was equipped with the sampling unit; the pumps were put into operation for one minute and then turned off again. The LV sampling units arrived at the vessel already fully assembled and were not re-opened. In contrast, the aluminum rings of the HV samplers were exposed to possible contamination from the vessel, as they were replaced in their respective samplers on deck. Operational blanks for both samplers were taken after transects 1, 4, and 7. The operational blanks were prepared and treated exactly like a sample, except with only a short pumping time (-1 min).

### Sample clean-up and procedural laboratory blanks

For prevention of secondary contamination during the sample clean-up, laboratory gear was made exclusively of glass, stainless-steel, or TEFLON®. The entire sample clean-up process was conducted under laminar flow in a safety cabinet (Claire pro B-2-130, Berner International GmbH, Germany) and with pre-filtered (glass fiber filter, 0.3 µm pore size, Whatman™; pre-treated in a muffle furnace at 500 °C for 4 h) chemicals and solutions only. Laboratory gear was cleaned with pre-filtered water and ethanol (96%) and kept covered with aluminum foil at all times. Sample clean-up in the laboratory was accompanied by full procedural blanks to monitor secondary contamination originating from the laboratory. The procedural blanks ($n = 6$ for LV samplers; $n = 12$ for HV samplers), were prepared in parallel to the samples in the laboratory. In the case of positive signals, the respective raw data of the blanks were subtracted from any sample raw data (based on derived peak area ratios calculated from signals divided by the internal standard signal of the pyrolysis process, c.f. 3.4). This avoided any over-quantification due to secondary contaminants introduced during analytical processing.

Low-volume (LV) samples. The stainless-steel meshes were transferred into beakers filled with ethanol (96%) and placed in an ultrasonic bath for 10 min to detach the particles from the mesh. Thereafter, the stainless-steel meshes were cleaned thoroughly with ethanol and the solutions, including the filter residues, were filtrated onto glass fiber filters (15 mm, 0.3 µm pore size, Whatman™; pre-treated in a muffle furnace at 500 °C for 4 h). The filter cakes on the glass fiber filters were rinsed with hydrogen peroxide (30% (v/v)) and petroleum ether to oxidize labile organic matter and to remove solvable lipids (e.g., waxes or paraffin) that might cause interferences during pyrolysis. Both chemicals remained on the filters for five minutes each to eliminate organic matter (hydrogen peroxide) and remove nonpolar components (petroleum ether). Thereafter the glass fiber filters including filter cakes were folded and transferred to stainless-steel pyrolysis cups (Eco Cups 80 LF, Frontier Labs, Japan).

High-volume (HV) samples. The aluminum rings were treated with an antistatic gun (Milty Pro Zerostat 3, Merck KGaA, Germany), to neutralize the static charge on the aluminum surfaces, and rinsed with ethanol to transfer the samples onto glass fiber filters (15 mm, 0.3 µm pore size, Whatman™; pre-treated in a muffle furnace at 500 °C for 4 h). Then, the glass fiber filters, including the filter cakes, were rinsed with hydrogen peroxide (30% (v/v)) and petroleum ether, folded, and placed in stainless-steel pyrolysis cups.

### Pyrolysis-GC/MS

Samples were analyzed with a micro-furnace pyrolyzer EGA/Py-3030D equipped with the auto-shot sampler AS−1020E (both FrontierLabs, Japan) was operated at 590 °C. The pyrolysis unit was coupled with an Agilent 6890N gas chromatograph linked to an Agilent MSD 5973 mass spectrometer. A deactivated retention gap in combination with a DB-5MS column was installed in the gas chromatograph. In the ion source, electron ionization was conducted at 230 °C with an ionization energy of 70 eV. Detailed information is listed in the supplementary information SI, (Table S7). This method has been successfully applied in previous publications[4,17,18,21,29,39].

Before measurement, 20 µL of deuterated polystyrene solution (dPS, Sigma Aldrich, Germany, 125 µg mL⁻¹ in dichloromethane), used as an internal pyrolysis process standard (ISTDpy) and 20 µL of tetra-methylammonium hydroxide solution (TMAH, 12.5% in methanol, Sigma Aldrich, Germany) for on-line derivatization and thermochemolysis were added to each pyrolysis cup[18,29].

### Polymer identification, quantification, and calibration

Identification, quantification, and calibration of the polymers C-PE, C-PS, C-PET, C-PMMA, C-PC, C-PA6, C-MDI-PUR, CTT, and TTT were conducted as described in former publications[18,21,29,39]. All the relevant details and information are provided in the supplementary information SI, (Tables S8, S9, and S10). The calibration functions were based on the peak area ratios of the respective polymer indicator ion relative to that of deuterated PS (dPS), used for internal standardization of the pyrolysis process (ISTDpy, $m/z$ 98, 2,4,6-Triphenyl-1-hexene). In contrast to the above-mentioned polymers calibrated in the range between 0.01 µg and 10 µg, C-PP was quantified via a 1-point-calibration (lowest calibration point, 0.85 µg), because the concentrations of the samples were always below the calibration range. The 1-point calibration was performed to obtain at least semi-quantitative data for C-PP. Before quantification, all sample raw data were corrected by the laboratory procedural blank.

As shown in a recent study by ref. 4, polymeric material of soot, e.g. from wood stove, released naphthalene during Py-GC/MS, even after sample treatment. Naphthalene is used as a C-PVC indicator, although highly unspecific, due to the absence of a more specific alternative. For terrestrial soot, a correction factor was introduced to calculate semi-quantitative C-PVC data. So far, no correction factor for interferences of marine soot has been determined. However, the first analysis of marine soot showed that the correction factor for terrestrial soot is not applicable to marine samples. Accordingly, in this study, C-PVC data are excluded from further discussions.

### Atmospheric transport modeling

For evaluation of potential MP sources, including the possible geographical and anthropogenic impacts of air masses arriving at

the deck of the R/V Heincke, two different atmospheric transport and dispersion models were applied: the FLEXPART and the HYSPLIT model.

**FLEXPLART model.** To track the origin of the air parcels and to generate emission footprints, the Lagrangian particle dispersion model, FLEXPART version 10.4[40], was used. The model was driven with 3-hourly operational meteorological wind fields retrieved from the European Centre for Medium-Range Weather Forecasts (ECMWF), consisting of 137 vertical levels and a horizontal resolution of 1° × 1°. In FLEXPART, computational particles were released at heights of 0–100 m from the moving receptor (R/V Heincke) every 4 h and were tracked 30 days backward in time in FLEXPART's retroplume mode.

The tracking includes gravitational settling for spherical particles of the size observed. FLEXPART differs from simple trajectory models due to its ability to simulate dry and wet deposition of gases or aerosols[41], turbulence[42], and unresolved mesoscale motions[43], while it includes a deep convection scheme[44]. For our simulations, we assumed that microplastics had a density of 1234 kg m$^{-3}$ as in ref. 8 and followed a logarithmic size distribution characterized by an aerodynamic mean diameter. In the present case, considering that particles <10 µm and >10 µm were detected, we used separate runs for 0.4, 3.0, 8.0, 10, 12, 18, and 25 µm.

The FLEXPART output consists of a gridded footprint emission sensitivity at a resolution of 0.5° × 0.5°. The emission sensitivity expresses the probability of any release occurring in any grid-cell to reach the receptor (R/V Heincke) during the 30-day particle tracking.

**HYSPLIT model.** Back trajectories were calculated using the HYSPLIT 4 model[45]. Meteorological input data were retrieved from the global National Centers for Environmental Prediction (NCEP)/National Center for Atmospheric Research (NCAR) reanalysis dataset[46]. The NCEP/NCAR reanalysis data provides 6-hourly data in 18 vertical levels with a horizontal resolution of 2.5° × 2.5°. The HYSPLIT model was run in the backward mode for periods of 24 h and 72 h, calculating the hourly latitudinal and longitudinal position of the air mass before arriving at R/V Heincke during sampling transects. The trajectory endpoint at R/V Heincke was set at a height of 30 m above sea level.

## Data availability
The MP data generated in this study are provided in the supplementary information. All FLEXPART version 10.4 simulation results are openly available and can be accessed from https://doi.org/10.5281/zenodo.7924016 or upon request to N.E. Additional HYSPLIT information can be obtained from the corresponding author.

## Code availability
The FLEXPART and HYSPLIT models are freely available to the scientific community and can be downloaded from https://www.flexpart.eu/ and https://www.ready.noaa.gov/HYSPLIT.php. Meteorological fields to run FLEXPART can be downloaded from ECMWF (https://www.ecmwf.int) following their terms/guidelines. For HYSPLIT meteorological data can be found on https://www.ready.noaa.gov/archives.php and in ref. 46. The maps for transects (Fig. 2a) and the HYSPLIT trajectories (Figs. 4b and S4b) were drawn in R using the "ggplot2" package world map data (map_data("world")). The ggplot citation is ref. 47. The FLEXPART footprints were plotted with Matlab 2022a (MATLAB. (2022). Version R2022a. Natick, Massachusetts: The MathWorks Inc.). All maps are included in the code.

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

## Acknowledgements

Sincere thanks to Anke Müllenmeister-Sawall for technical assistance, and the captain and crew of the R/V Heincke (HE578). Furthermore, we highly appreciate the language check provided by Cleo L. Davie-Martin. This study was funded by the German Federal Ministry of Education and Research (Bundesministerium für Bildung und Forschung, BMBF), in the joint research project FACTS (grant IDs 03F0849C (I.G., B.M.S.B., O.W., and J.S.), 03F0849D (A.H.); 03F0849A (G.G.); JPI-Oceans). Furthermore, thanks to the Norwegian Research Council (NRC), for funding PlastPoll21 (grant ID 322191 (I.G., D.H., and V.N.)) and FACTS (grant ID 311316 (N.E., D.H., and S.E.)). Funding by the Deutsche Forschungsgemeinschaft (DFG, German Research Foundation) – project number 391977956 (C.G.) – SFB1357 is also gratefully acknowledged. FLEXPART analyses were also funded by ATMO-ACCESS, EU grant agreement No 101008004. All authors would like to thank two anonymous reviewers and Sally Gaw for their thoroughly and constructive reviews.

## Author contributions

The manuscript was written through contributions of all authors. All authors have given approval to the final version of the manuscript. I.G. performed laboratory work, the majority of measurements with Py-GC/MS, data analysis, wrote the main part of the manuscript, and produced figures, tables, and supplementary material. D.H., V.N., A.H., and G.G. organized the sampling campaign, edited, and approved the manuscript. J.S. conducted laboratory work, performed measurements with Py-GC/MS, and data analysis. C.G. conducted the sampling campaign, and performed the HYSPLIT modeling. N.E. and S.E. performed the FLEXPART modeling, edited, and approved the manuscript. O.W. gave feedback, edited, and approved the manuscript. B.S.-B. supervised all stages of the project, performed data analysis, contributed valuable discussion, edited, and approved the manuscript.

## Funding

## Competing interests

The authors declare no competing interests.
