## [Peer Review File · Nature Communications]

Occurrence and backtracking of microplastic mass loads including tire wear particles in northern Atlantic airReviewer #1 (Remarks to the Author):

The work by Goßmann et al., on the Occurrence and backtracking of microplastic mass loads including tire wear particles in Northern Atlantic air was reviewed. The authors measured MP composition and MP mass loads in the marine atmosphere with samples collected with active air sampling devices (low- and high-volume samplers) and analysis via a Pyr-GC/MS. The determination of MPs in air is quite interesting and there is a lack of mass data for reliable estimations of the contribution of this emission pathway to the overall MP loads. The manuscript is well written and carries novel information and data set. The results and discussion are supported by the methodologies.

I have only one minor comment.

Lines 304 and 310: what was the purpose or reason of using petroleum ether. Did the petroleum ether produced any interfering compounds or makers during pyrolysis. Was this checked? How was this used? Was it added directly onto the filters on the filtration system or units after the samples have first being filtered? If this was used to degrade organic materials was this proven with experiments? I have also same concern for the hydrogen peroxide, how was this added. Maybe add some details in the SI.

Reviewer #2 (Remarks to the Author):

The present manuscript reports an innovative study that collected marine atmospheric microplastics. Two different air samplers were used and polymer clusters were identified and quantified through Py-GC/MS. Moreover, atmospheric transport and dispersion models were applied to correlate the results to possible sources of microplastic pollution. I particular appreciated the operational blank part that was thorough and well designed. Cross-contamination is an important issue in microplastic collection and investigation, particularly when the sampling is performs on ships. The research work is consistent and well designed and could pave the way to the development of standardised procedures to collect and compare marine atmospheric microplastics. However, in the current form, the work is sometimes difficult to follow and results are not presented in a clear and straightforward way. I recommend to address the following issues prior to publication in Nature Communications.

1) Results and Discussion: this part need to be rewritten taking into account that the method part is at the end of the article. In the current form, it is difficult to follow the discussion because it takes for granted the knowledge of the method part (see for example lines 92-93).

2) Why only polyamide 6 was considered among the different polyamides?

3) Fig. 2 should be bigger and the colours in 2a cannot be easily distinguished. In 2c-d please use different colours for C-MDI-PUR e Car tire tread, the present ones are not easily distinguished.

4) Lines 129-143: the discussion here is based on clusters C-PS, C-PC, C-PP and C-PET that are affected by contamination. These needs to be clearly stated and taken into account in the discussion of the results. please add that C-PS could be affected by contamination.

5) Besides Fig. S3, were other samples observed by optical microscopy before the Py-GC/MS? Just to record the form of the particles found. For example, the high concentration of C-PET clusters could be due to fibres?

6) The sampling time can differ from 12h to 91h. This is quite a big range. Was the sampling time taken into account when considering concentration of MP found and differences among samples? This is an important parameter that can influence the measurement and need to be discussed in the article and considered for future developments of standardized methods.

7) In the methods section, it is stated that HV samplers were deployed in duplicate. In the SI, values of each duplicate need to be included. Was LV samplers deployed also in duplicate?

8) Please adjust the reference citation style in the text.

Reviewer #3 (Remarks to the Author):

This is an important study as to date few studies have compared results for microplastics using different types of air sampling. The care taken in this study to minimise contamination of the samples is exemplary. The methods were provided in enough detail for the work to be reported.

This study also combined field data with modelling to identify potential sources.

This study was one of the first to use pyrolysis GC-MS to identify the polymer types in air samples and the first for marine atmospheric samples. The approach of size fractionation during sample collection coupled with pyrolysis GC-MS enabled the detection of <10 um microplastics.

The key findings from the research were that microplastics may be being released from the marine environment back into the atmosphere. Until recently the line of thinking was that the oceans were a sink rather than a potential source of microplastics.

The conclusions are consistent with the data and data analysis.

Sally Gaw

REVIEWERS' COMMENTS

Answers to the reviewers are indented and given in blue colored writing. New text sections or paragraphs are given in italic writing from the track changes version, so it is clear what exactly has been changed. The respective lines for the newly added information are referring to the lines in the track-changes (TC) manuscript.

Reviewer #1 (Remarks to the Author):

The work by Goßmann et al., on the Occurrence and backtracking of microplastic mass loads including tire wear particles in Northern Atlantic air was reviewed. The authors measured MP composition and MP mass loads in the marine atmosphere with samples collected with active air sampling devices (low- and high-volume samplers) and analysis via a Pyr-GC/MS. The determination of MPs in air is quite interesting and there is a lack of mass data for reliable estimations of the contribution of this emission pathway to the overall MP loads. The manuscript is well written and carries novel information and data set. The results and discussion are supported by the methodologies.

Dear reviewer, thank you very much for the time to review this manuscript and for the kind feedback! It is highly appreciated.

I have only one minor comment.

Lines 304 and 310: what was the purpose or reason of using petroleum ether. Did the petroleum ether produced any interfering compounds or makers during pyrolysis. Was this checked? How was this used? Was it added directly onto the filters on the filtration system or units after the samples have first being filtered? If this was used to degrade organic materials was this proven with experiments? I have also same concern for the hydrogen peroxide, how was this added. Maybe add some details in the SI.

Thank you very much for the comment. In the method section it was indeed not clarified why hydrogen peroxide and petroleum ether were added and especially that it was applied to both sample types. Thank you very much for pointing this out. We do not have any interferences in the Py-GC/MS with either of the chemicals. Hydrogen peroxide is an integral part of our sample clean-up. It is an integral oxidation and necessary step to get labile organic material removed. The intention of using petroleum ether is to remove any organic solvable lipids (e.g. wax or paraffin particles) prior to pyrolysis. Both steps are intended to prevent interferences in Py-GC/MS as much as possible. We have documented this in our previous publications. To clarify the use of both reagents we rephrased the respective paragraphs as follows:

“The filter cakes on the glass fiber filters were rinsed with hydrogen peroxide (30% (v/v)) and petroleum ether to oxidize labile organic matter and to remove solvable lipids (e.g., waxes or paraffin) that might cause interferences during pyrolysis. Both chemicals remained on the filter for five minutes each to eliminate organic matter (hydrogen peroxide) and remove nonpolar components (petroleum ether). Thereafter the glass fiber filter including filter cakes were folded, and transferred to stainless-steel pyrolysis cups (Eco Cups 80 LF, Frontier Labs, Japan).” (line 351 – 357 (TC))

“Then, the glass fiber filters, including the filter cakes, were rinsed ~~washed~~ with ~~petroleum ether and hydrogen peroxide (30% (v/v)) and petroleum ether~~, folded, and placed in stainless-steel pyrolysis cups.” (line 361 – 363 (TC))

Reviewer #2 (Remarks to the Author):

The present manuscript reports an innovative study that collected marine atmospheric microplastics. Two different air samplers were used and polymer clusters were identified and quantified through Py-GC/MS. Moreover, atmospheric transport and dispersion models were applied to correlate the results to possible sources of microplastic pollution. I particularly appreciated the operational blank part that was thorough and well designed. Cross-contamination is an important issue in microplastic collection and investigation, particularly when the sampling is performed on ships. The research work is consistent and well designed and could pave the way to the development of standardised procedures to collect and compare marine atmospheric microplastics. However, in the current form, the work is sometimes difficult to follow and results are not presented in a clear and straightforward way. I recommend to address the following issues prior to publication in Nature Communications.

Dear reviewer, thank you very much for taking the time to review our manuscript. Thank you very much for your feedback and especially for pointing out the blank discussion. We also think that representative operational/field and lab blanks are the fundamentals for a reliable MP analysis and quantification and tried our best to underline its importance and also the difficulties that occur within this subject.

1) Results and Discussion: this part needs to be rewritten taking into account that the method part is at the end of the article. In the current form, it is difficult to follow the discussion because it takes for granted the knowledge of the method part (see for example lines 92-93).

Thank you very much for pointing this out! Indeed, we prepared the manuscript in the order for introduction, methods and then results and discussions. We adapted the text sections accordingly. We removed phrases as “as already mentioned” and explained abbreviations, which were originally first mentioned in the methods sections. We hope, it is clearer now!

“To minimize secondary contamination from the ship in advance, the sampling devices were positioned at elevated sites at the ship’s bow and sampling was strictly restricted to steaming phases only. However, ~~o~~Operational blanks for air sampling were taken throughout the entire sampling period to monitor secondary contamination released from the ship’s environment or the two different sampling devices. ~~As already mentioned,~~ Low-volume (LV, 54 to 417 m³ air per sample) samplers were pre-assembled beforehand under a laboratory clean bench. In contrast, high-volume (HV, 288 to 2184 m³ air per sample) samplers were regularly opened on board ~~regularly~~ to exchange the aluminum rings used as sampling targets.” (line 99 – 106 (TC))

“In the following, the polymer clusters found in the respective operational blanks are discussed and their secondary contamination with its potential effects on further quantification is evaluated.” (line 107 – 109 (TC))

“The indicator ion for C-PMMA appeared almost ~~omnipresent~~ ubiquitous in ~~all LV and HV sample~~ the seven transects (T1 – T7) in both samplers (LV and HV) and all investigated size fractions (> 10 µm for LV & HV samples; 5 -10 µm for LV samples).” (line 110 – 112 (TC))

“Total mass loads ranged from < limit of quantification (LOQ) to 1.82 ng MP m⁻³. Limits of detection (LOD) and ~~quantification~~ (LOQ) are displayed in the supplementary information (SI, Table S2).” (line 144 – 146 (TC))

“The seven published, particle-number based studies mentioned in the introduction and the supplement (SI, Table S1) did not include TWP.” (line 168 – 169 (TC))

“To discover potential MP sources in the marine atmosphere, we used the Hybrid Single-Particle Lagrangian Integrated Trajectory (HYSPLIT) and the FLExible PARTicle (FLEXPART) dispersion models ~~were consulted~~ to obtain information about

the origin of air masses, which arrived to the ship and were hence sampled.” (line 234 – 237 (TC))

2) Why only polyamide 6 was considered among the different polyamides?

Thank you for the comment. To analyse a polymer or its respective cluster with Py-GC/MS we need the pure polymer standard and a respective indicator compound as a thermally generated volatile decomposition product, which ideally exclusively indicated the presence of the respective polymer. Therefore, we only include polyamide 6 and polyamide 66, as the most representative polyamides into our analysis. So far, PA66 was rarely detected in the environmental samples analysed. Accordingly, we focussed on PA6 here. Since our measurements are conducted in full scan mode and contain a known concentration of internal standard a retrospective analysis of further polymers is possible if desired and considered useful.

3) Fig. 2 should be bigger and the colours in 2a cannot be easily distinguished. In 2c-d please use different colours for C-MDI-PUR e Car tire tread, the present ones are not easily distinguished.

Thank you for the advice. We enlarged Figure 2. We added the name of the transects right next to the respective transect to make it more comprehensible. Furthermore, we added cross hatching to certain polymers (C-PC and C-MDI-PUR). Now the figures are readable for people with red/green-blindness and C-MDI-PUR and CTT are distinguishable.

4) Lines 129-143: the discussion here is based on clusters C-PS, C-PC, C-PP and C-PET that are affected by contamination. These needs to be clearly stated and taken into account in the discussion of the results. please add that C-PS could be affected by contamination.

Thank you, this is a very good idea to make this even clearer in the text. Therefore, we decided to additionally label the impaired polymer cluster not only in the figures, but also in the text with the same symbol.

“Instead, these ~~respective~~ polymers clusters are marked (Δ) in the respective figures and Δ in the text to indicate that the concentration might be partly impaired by secondary contamination.” (line 136 – 138 (TC))

We noticed a mistake in section 2.1., we wrote that both C-PS and C-PET appeared occasionally in the operational blanks of both samplers, which was wrong. In the respective figures and in the supplement with the quantitative data tables however, it was displayed correctly. We corrected the respective phrase accordingly:

“C-PS was occasionally detected in ~~and C-PET for~~ both samplers, as well as ~~and C-PP and C-PET for the HV sampler were detected occasionally~~, but did not show any operational blank related pattern and thus, did not suggest an overall secondary contamination during sampling or sample preparation.” (line 130 – 133 (TC))

5) Besides Fig. S3, were other samples observed by optical microscopy before the Py-GC/MS? Just to record the form of the particles found. For example, the high concentration of C-PET clusters could be due to fibres?

Thank you for the question. We documented all filter cakes and on many of them fibers were visible. However not as extreme as in Fig S3.

6) The sampling time can differ from 12h to 91h. This is quite a big range. Was the sampling time taken into account when considering concentration of MP found and differences among

samples? This is an important parameter that can influence the measurement and need to be discussed in the article and considered for future developments of standardized methods.

Thank you for this comment. The sampling time and the sample volumes differed quite a lot depending on the length of the driven transect. Due to the broad range of volumes, we took that into account with respect to the operational blank discussion while referring to sample⁻¹ and not ng m⁻³ (see section 2.1). To highlight these volume differences, the total sample volumes were also included in the above-mentioned section:

“However, operational blanks for air sampling were taken throughout the entire sampling period to monitor secondary contamination released from the ship’s environment or the two different sampling devices. As already mentioned, Low-volume (LV, 54 to 417 m³ air per sample) samplers were pre-assembled beforehand under a laboratory clean bench. In contrast, high-volume (HV, 288 to 2184 m³ air per sample) samplers were regularly opened on board regularly to exchange the aluminum rings used as sampling targets.” (line 101 – 106 (TC))

We have tried to emphasize clearly the different LODs and LOQs of polymers in the discussion and have included the respective values for the polymers analysed here in the supplement (SI, Table S2). The respective paragraph was rephrased accordingly:

“Other polymer clusters might have also been present in the LV samples as well, but due to higher LODs of some polymers (e.g., CTT), they might have evaded identification in the comparably low sample volumes.” (line 183 – 185 (TC))

Furthermore, a paragraph was added in the discussion comparing the LV and HV sampler:

“In contrast, the elevated air volumes of the HV sampler revealed a greater polymer cluster diversity in the air. Larger sample volumes often ensure a more reliable analysis of polymers with higher LOD and LOQ, which might directly result in a higher polymer cluster diversity for the HV samples compared to the LV samples.” (line 221 – 224 (TC))

7) In the methods section, it is stated that HV samplers were deployed in duplicate. In the SI, values of each duplicate need to be included. Was LV samplers deployed also in duplicate?

Thank you for the advice. We adapted Table S4 accordingly with the quantitative HV results and included the individual values plus the mean values. The LV samples were not deployed as duplicates.

8) Please adjust the reference citation style in the text.

Citations and reference lists were created with Mendeley and the citation style chosen is from *Nature*. Therefore, we are not entirely sure, how you would like the references to be adjusted.

Reviewer #3 (Remarks to the Author):

This is an important study as to date few studies have compared results for microplastics using different types of air sampling. The care taken in this study to minimise contamination of the samples is exemplary. The methods were provided in enough detail for the work to be reported.

This study also combined field data with modelling to identify potential sources.

This study was one of the first to use pyrolysis GC-MS to identify the polymer types in air samples and the first for marine atmospheric samples. The approach of size fractionation

during sample collection coupled with pyrolysis GC-MS enabled the detection of <10 um microplastics.

The key findings from the research were that microplastics may be being released from the marine environment back into the atmosphere. Until recently the line of thinking was that the oceans were a sink rather than a potential source of microplastics.

The conclusions are consistent with the data and data analysis.

Sally Gaw

Dear Sally Gaw, thank you very much for reviewing our manuscript. Your effort is greatly appreciated. Also thank you very much for your kind feedback.